# SEMANTICS PRESERVING ADVERSARIAL ATTACKS

## ABSTRACT

While progress has been made in crafting visually imperceptible adversarial examples, constructing semantically meaningful ones remains a challenge. In this paper, we propose a framework to generate semantics preserving adversarial examples. First, we present a manifold learning method to capture the semantics of the inputs. The motivating principle is to learn the low-dimensional geometric summaries of the inputs via statistical inference. Then, we perturb the elements of the learned manifold using the Gram-Schmidt process to induce the perturbed elements to remain in the manifold. To produce adversarial examples, we propose an efficient algorithm whereby we leverage the semantics of the inputs as a source of knowledge upon which we impose adversarial constraints. We apply our approach on toy data, images and text, and show its effectiveness in producing semantics preserving adversarial examples which evade existing defenses against adversarial attacks.

## 1 INTRODUCTION

In response to the susceptibility of deep neural networks to small adversarial perturbations (Szegedy et al., 2014), several defenses have been proposed (Liu et al., 2019; Sinha et al., 2018; Raghunathan et al., 2018; Madry et al., 2017; Kolter & Wong, 2017). Recent attacks have, however, cast serious doubts on the robustness of these defenses (Athalye et al., 2018; Carlini & Wagner, 2016). A standard way to increase robustness is to inject adversarial examples into the training inputs (Goodfellow et al., 2014a). This method, known as adversarial training, is however sensitive to distributional shifts between the inputs and their adversarial examples (Ilyas et al., 2019). Indeed, distortions, occlusions or changes of illumination in an image, to name a few, do not always preserve the nature of the image. In text, slight changes to a sentence often alter its readability or lead to substantial differences in meaning. Constructing semantics preserving adversarial examples would provide reliable adversarial training signals to robustify deep learning models, and make them generalize better. However, several approaches in adversarial attacks fail to enforce the semantic relatedness that ought to exist between the inputs and their adversarial counterparts. This is due to inadequate characterizations of the semantics of the inputs and the adversarial examples — Song et al. (2018) and Zhao et al. (2018b) confine the distribution of the latents of the adversarial examples to a Gaussian. Moreover, the search for adversarial examples is customarily restricted to uniformly-bounded regions or conducted along suboptimal gradient directions (Szegedy et al., 2014; Kurakin et al., 2016; Goodfellow et al., 2014b).

In this study, we introduce a method to address the limitations of previous approaches by constructing adversarial examples that explicitly preserve the semantics of the inputs. We achieve this by characterizing and aligning the low dimensional geometric summaries of the inputs and the adversarial examples. The summaries capture the semantics of the inputs and the adversarial examples. The alignment ensures that the adversarial examples reflect the unbiased semantics of the inputs. We decompose our attack mechanism into: (i.) *manifold learning*, (ii.) *perturbation invariance*, and (iii.) *adversarial attack*. The motivating principle behind step (i.) is to learn the low dimensional geometric summaries of the inputs via statistical inference. Thus, we present a variational inference technique that relaxes the rigid Gaussian prior assumption typically placed on VAEs encoder networks (Kingma & Welling, 2014) to capture faithfully such summaries. In step (ii.), we develop an approach around the *manifold invariance concept* of (Roussel, 2019) to perturb the elements of the learned manifold while ensuring the perturbed elements remain within the manifold. Finally, in step (iii.), we propose a learning algorithm whereby we leverage the rich semantics of the inputs and the perturbations as a source of knowledge upon which we impose adversarial constraints to produce adversarial examples. Unlike (Song et al., 2018; Carlini & Wagner, 2016; Zhao et al., 2018b; Goodfellow et al., 2014b) that resort to a costly search of adversarial examples, our algorithm is efficient and end-to-end.

The main contributions of our work are thus: (i.) a variational inference method for manifold learning in the presence of continuous latent variables with minimal assumptions about their distribution, (ii.) an intuitive perturbation strategy that encourages perturbed elements of a manifold to remain within the manifold, (iii.) an end-to-end and computationally efficient algorithm that combines (i.) and (ii.) to generate adversarial examples in a black-box setting, and (iv.) illustration on toy data, images and text, as well as empirical validation against strong certified and non-certified adversarial defenses.

## 2 PRELIMINARIES & ARCHITECTURE

**Notations.** Let $x$ be a sample from the input space $\mathcal{X}$, with label $y$ from a set of possible labels $\mathcal{Y}$, and $\mathcal{D} = \{x_n\}_{n=1}^N$ a set of $N$ such samples $x$. Also, let $d$ be a distance measure on $\mathcal{X}$ capturing closeness in input space, or on $\mathcal{Z}$, the embedding space of $\mathcal{X}$, capturing semantics similarity.

**Adversarial Examples.** Given a classifier $g$, and its loss function $\ell$, an adversarial example of $x$ is produced by maximizing the objective below over an $\epsilon$-radius ball around $x$ (Athalye et al., 2017).

$$x' = \arg\max_{x' \in \mathcal{X}} \ell(g(x'), y) \text{ such that } x' \in \mathcal{B}(x; \epsilon)$$

Above, the search region for adversarial examples is confined to a uniformly-bounded ball $\mathcal{B}(x; \epsilon)$. In reality, however, the shape imposed on $\mathcal{B}$ is quite restrictive as the optimal search region may have a different topology. It is also common practice to produce adversarial examples in the input space $\mathcal{X}$ — via an exhaustive and costly search procedure (Shaham et al., 2018; Song et al., 2018; Zhao et al., 2018b; Athalye et al., 2017; Carlini & Wagner, 2016; Goodfellow et al., 2014b). Unlike these approaches, however, we wish to operate in $\mathcal{Z}$, the lower dimensional embedding space of $\mathcal{X}$, with minimal computational overhead. Our primary intuition is that $\mathcal{Z}$ *captures well the semantics of* $\mathcal{D}$.

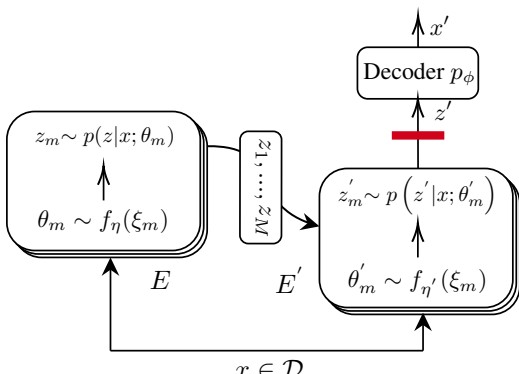

**Attack Model.** Given a sample $x \in \mathcal{D}$ and its class $y \in \mathcal{Y}$, we want to construct an adversarial example $x'$ that shares the same semantics as $x$. We assume the semantics of $x$ (resp. $x'$) is modeled by a learned latent variable model $p(z)$ (resp. $p'(z')$), where $z, z' \in \mathcal{Z}$. In this setting, observing $x$ (resp. $x'$) is conditioned on the observation model $p(x|z)$ (resp. $p(x'|z')$), that is: $x \sim p(x|z)$ and $x' \sim p(x'|z')$, with $z \sim p(z)$ and $z' \sim p'(z')$. We learn this model in a way that $d(x, x')$ is small and $g(x) = y \wedge g(x') \neq y$ while ensuring also that $d(z, z')$ is small.

Intuitively, we get the latent $z \sim p(z)$ which encodes the semantics of $x$. Then, we perturb $z$ in a way that its perturbed version $z' \sim p'(z')$ lies in the manifold that supports $p(z)$. We define a manifold as a set of points in $\mathcal{Z}$ where every point is locally Euclidean (Roussel, 2019). We devise our perturbation procedure by generalizing the *manifold invariance concept* of (Roussel,

Figure 1: Architecture. The set of model parameters $\Theta = \{\theta_m\}_{m=1}^M$ and $\Theta' = \{\theta'_m\}_{m=1}^M$ are sampled from the recognition networks $f_\eta$ and $f_{\eta'}$. Given an input $x \in \mathcal{D}$, we use $E$ to sample the latent codes $z_1, ..., z_M$ via $\Theta$. These codes are passed to $E'$ to learn their perturbed versions $z'_1, ..., z'_M$ using $\Theta'$. The output $x' \sim p_\phi(x'|z')$ is generated via posterior sampling of a $z'$ (in red).

2019) to $\mathcal{Z}$. For that, we consider two embedding maps $h \colon \mathcal{X} \to \mathcal{Z}$ and $h' \colon \mathcal{X} \to \mathcal{Z}$, parameterized by $\theta$ and $\theta'$, as surrogates for $p(z)$ and $p(z')$. We assume $\theta$ and $\theta'$ follow the implicit distributions $p(\theta)$ and $p(\theta')$. In the following, we consider $M$ such embedding maps $h$ and $h'$.[1] If we let $z = h(x; \theta)$ and $z' = h'(x; \theta')$, we ensure that $z'$ is in the manifold that supports $p(z)$ by constraining $d(z, z')$ to be small. Then, given a map $dec_\phi \colon \mathcal{Z} \to \mathcal{X}$, we craft $x' = dec_\phi(z')$ in a way that $d(x, x')$ is small and $g(x) = y \wedge g(x') \neq y$. Then, we say that $x'$ is adversarial to $x$ and preserves its semantics.

**Model Architecture.** To implement our attack model, we propose as a framework the architecture illustrated in Figure 1. Our framework is essentially a variational auto-encoder with two encoders $E$ and $E'$ that learn the geometric summaries of $\mathcal{D}$ via statistical inference. We present two inference mechanisms — *implicit manifold learning via Stein variational gradient descent* (Liu & Wang, 2016) and *Gram-Schmidt basis sign method* (Dukes, 2014) — to draw instances of model parameters from

---

[1]The reason why we consider $M$ instances of $h$ and $h'$ will become apparent in Section 3.

the implicit distributions $p(\theta)$ and $p(\theta')$ that we parameterize $E$ and $E'$ with. Both encoders optimize the uncertainty inherent to embedding $\mathcal{D}$ in $\mathcal{Z}$ while guaranteeing easy sampling via Bayesian ensembling. Finally, the decoder $p_\phi$ acts as a generative model for constructing adversarial examples.

**Threat Model.** We consider in this paper a *black-box* scenario where we, as an attacker, have only access to the predictions of a classifier $g$. As the attacker, we want to construct adversarial examples not knowing the intricacies of $g$ such as its loss function, nor having access to its gradient. We focus on this scenario because it is challenging and more plausible in real-life than the white-box case. This threat model serves to evaluate both certified defenses and non-certified ones under our attack model

## 3 IMPLICIT MANIFOLD LEARNING

Manifold learning is based on the assumption that high dimensional data lies on or near lower dimensional manifolds in a data embedding space. In the variational auto-encoder (VAE) (Kingma & Welling, 2014) setting, the datapoints $x_n \in \mathcal{D}$ are modeled via a decoder $x_n \sim p(x_n|z_n; \phi)$. To learn the parameters $\phi$, one typically maximizes a variational approximation to the empirical expected log-likelihood $1/N \sum_{n=1}^N \log p(x_n; \phi)$, called evidence lower bound (ELBO), defined as:

$$\mathcal{L}_e(\phi, \psi; x) = \mathbb{E}_{z|x;\psi} \log \left[ \frac{p(x|z; \phi)p(z)}{q(z|x; \psi)} \right] = -\mathbb{KL}(q(z|x; \psi)\|p(z|x; \phi)) + \log p(x; \phi). \quad (1)$$

The expectation $\mathbb{E}_{z|x;\psi}$ can be re-expressed as a sum of a reconstruction loss, or expected negative log-likelihood of $x$, and a $\mathbb{KL}(q(z|x; \psi)\|p(z))$ term. The $\mathbb{KL}$ term acts as a regularizer and forces the encoder $q(z|x; \psi)$ to follow a distribution similar to $p(z)$. In VAEs, $p(z)$ is defined as a *spherical Gaussian*. That is, VAEs learn an encoding function that maps the data manifold to an isotropic Gaussian. However, Jimenez Rezende & Mohamed (2015) have shown that the Gaussian form imposed on $p(z)$ leads to uninformative latent codes; *hence to poorly learning the semantics of $\mathcal{D}$* (Zhao et al., 2017). To sidestep this issue, we minimize the divergence term $\mathbb{KL}(q(z|x; \psi)\|p(z|x; \phi))$ using Stein Variational Gradient Descent (Liu & Wang, 2016) instead of explicitly optimizing the ELBO.

**Stein Variational Gradient Descent (SVGD)** is a nonparametric variational inference method that combines the advantages of MCMC sampling and variational inference. Unlike ELBO (Kingma & Welling, 2014), SVGD does not confine a target distribution $p(z)$ it approximates to simple or tractable parametric distributions. It remains yet an efficient algorithm. To approximate $p(z)$, SVGD maintains $M$ particles $\mathbf{z} = \{z_m\}_{m=1}^M$, initially sampled from a simple distribution, it iteratively transports via functional gradient descent. At iteration $t$, each particle $z_t \in \mathbf{z}_t$ is updated as follows:

$$z_{t+1} \leftarrow z_t + \alpha_t \tau(z_t) \text{ where } \tau(z_t) = \frac{1}{M} \sum_{m=1}^M \left[ k(z_t^m, z_t) \nabla_{z_t^m} \log p(z_t^m) + \nabla_{z_t^m} k(z_t^m, z_t) \right],$$

where $\alpha_t$ is a step-size and $k(.,.)$ is a positive-definite kernel. In the equation above, each particle determines its update direction by consulting with other particles and asking their gradients. The importance of the latter particles is weighted according to the distance measure $k(.,.)$. Closer particles are given higher consideration than those lying further away. The term $\nabla_{z^m} k(z^m, z)$ is a regularizer that acts as a repulsive force between the particles to prevent them from collapsing into one particle. Upon convergence, the particles $z_m$ will be unbiased samples of the true implicit distribution $p(z)$.

**Manifold Learning via SVGD.** To characterize the manifold of $\mathcal{D}$, which we denote $\mathcal{M}$, we learn the encoding function $q(.; \psi)$. Similar to Pu et al. (2017), we optimize the divergence $\mathbb{KL}(q(z|x; \psi)\|p(z|x; \phi))$ using SVGD. As an MCMC method, SVGD, however, induces inherent uncertainty we ought to capture in order to learn $\mathcal{M}$ efficiently. To potentially capture such uncertainty, Pu et al. (2017) use dropout. However, according to Hron et al. (2017), dropout is not principled. Bayesian methods, on the contrary, provide a principled way to model uncertainty through the posterior distribution over model parameters. Kim et al. (2018) have shown that SVGD can be cast as a Bayesian approach for parameter estimation and uncertainty quantification. Since SVGD always maintains $M$ particles, we introduce thus $M$ instances of model parameters $\Theta = \{\theta_m\}_{m=1}^M$, where every $\theta_m \in \Theta$ is a particle that defines the weights and biases of a Bayesian neural network.

For large $M$, however, maintaining $\Theta$ can be computationally prohibitive because of the memory footprint. Furthermore, the need to generate the particles during inference for each test case is undesirable. To sidestep these issues, we maintain only one (recognition) network $f_\eta$ that takes as

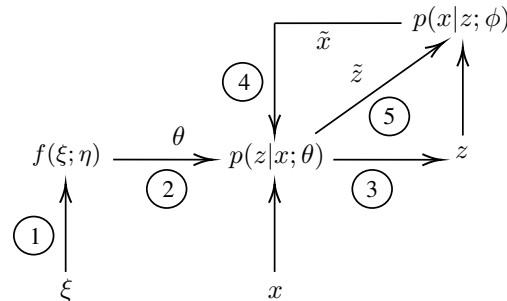

**Algorithm 1** Inversion with one particle $\boldsymbol{\theta}$.

**Require:** Input $x \in \mathcal{D}$
**Require:** Model parameters $\boldsymbol{\eta}$
1: Sample $\boldsymbol{\xi} \sim \mathcal{N}(\mathbf{0}, \mathbf{I})$
2: Sample $\boldsymbol{\theta} \sim \boldsymbol{f_\eta}(\boldsymbol{\xi})$
3: Given $x$, sample $z \sim p(z|x; \boldsymbol{\theta})$
4: Sample $\tilde{x} \sim p(x|z, \boldsymbol{\phi})$
5: Sample $\tilde{z} \sim p(z|\tilde{x}, \boldsymbol{\theta})$
6: Use $x$ and $\tilde{z}$ to compute $p(\tilde{z}|x; \boldsymbol{\theta})$

Figure 2: Inversion. Process for computing the likelihood $p(\mathcal{D}|\theta)$. As the decoder $p_\phi$ gets accurate, the error $\|x - \tilde{x}\|_2$ becomes small (see Algorithm 2), and we get closer to sampling the optimal $\tilde{z}$.

input $\xi_m \sim \mathcal{N}(\mathbf{0}, \mathbf{I})$ and outputs a particle $\theta_m$. The recognition network $f_\eta$ learns the trajectories of the particles as they get updated via SVGD. $f_\eta$ serves as a proxy to SVGD sampling strategy, and is refined through a small number of gradient steps to get good generalization.

$$\eta^{t+1} \leftarrow \arg\min_\eta \sum_{m=1}^M \left\| \underbrace{f(\xi_m; \eta^t)}_{\theta_m^t} - \theta_m^{t+1} \right\|_2 \quad \text{with } \theta_m^{t+1} \leftarrow \theta_m^t + \alpha_t \tau(\theta_m^t),$$

$$\text{where} \quad \tau(\theta^t) = \frac{1}{M} \sum_{j=1}^M \left[ k(\theta_j^t, \theta^t) \nabla_{\theta_t^j} \log p(\theta_j^t) + \nabla_{\theta_t^j} k(\theta_j^t, \theta^t) \right]. \tag{2}$$

We use the notation $\text{SVGD}_\tau(\Theta)$ to denote an SVGD update of $\Theta$ using the operator $\tau(.)$. As the particles $\theta$ are Bayesian, upon observing $\mathcal{D}$, we update the prior $p(\theta_j^t)$ to obtain the posterior $p(\theta_j^t|\mathcal{D}) \propto p(\mathcal{D}|\theta_j^t)p(\theta_j^t)$ which captures the uncertainty. We refer the reader to Appendix A for a formulation of $p(\theta_j^t|\mathcal{D})$ and $p(\mathcal{D}|\theta_j^t)$. The data likelihood $p(\mathcal{D}|\theta_j^t)$ is evaluated over all pairs $(x, \tilde{z})$ where $x \in \mathcal{D}$ and $\tilde{z}$ is a dependent variable. However, $\tilde{z}$ is not given. Thus, we introduce the *inversion process* described in Figure 2 to generate such $\tilde{z}$ using Algorithm 1. For any input $x \in \mathcal{D}$, we sample its latent code $z$ from $p(z|x; \mathcal{D})$, which we approximate by Monte Carlo over $\Theta$; that is:

$$p(z|x; \mathcal{D}) = \int p(z|x; \theta)p(\theta|\mathcal{D})dz \approx \frac{1}{M} \sum_{m=1}^M p(z|x; \theta_m) \text{ where } \theta_m \sim p(\theta|\mathcal{D}). \tag{3}$$

## 4 PERTURBATION INVARIANCE

Here, we focus on perturbing the elements of $\mathcal{M}$. We want the perturbed elements to reside in $\mathcal{M}$ and exhibit the semantics of $\mathcal{D}$ that $\mathcal{M}$ captures. Formally, we seek a linear mapping $h': \mathcal{M} \to \mathcal{M}$ such that for any point $z \in \mathcal{M}$, a neighborhood $\mathcal{U}$ of $z$ is *invariant* under $h'$; that is: $z' \in \mathcal{U} \Rightarrow h'(z') \in \mathcal{U}$. In this case, we say that $\mathcal{M}$ is *preserved under* $h'$. Trivial examples of such mappings are linear combinations of the basis vectors of subspaces $\mathcal{S}$ of $\mathcal{M}$ called linear spans of $\mathcal{S}$.

Rather than finding a linear span $h'$ directly, we introduce a new set of instances of model parameters $\Theta' = \{\theta'_m\}_{m=1}^M$. Each $\theta'_m$ denotes the weights and biases of a Bayesian neural network. Then, for any input $x \in \mathcal{D}$ and its latent code $z \sim p(z|x; \mathcal{D})$, a point in $\mathcal{M}$, we set $h'(z) = z'$ where $z' \sim p(z'|x; \mathcal{D})$. We approximate $p(z'|x; \mathcal{D})$ by Monte Carlo using $\Theta'$, as in Equation 3. We leverage the local smoothness of $\mathcal{M}$ to learn each $\theta'_m$ in a way to encourage $z'$ to *reside in $\mathcal{M}$ in a close neighborhood of* $z$ using a technique called Gram-Schmidt Basis Sign Method.

**Gram-Schmidt Basis Sign Method (GBSM).** Let $\mathbf{X}$ be a batch of samples of $\mathcal{D}$, $\mathbf{Z}_m$ a set of latent codes $z_m \sim p(z|x; \theta_m)$ where $x \in \mathbf{X}$, and $\theta_m \in \Theta$. For any $m \in \{1.., M\}$, we learn $\theta'_m$ to generate perturbed versions of $z_m \in \mathbf{Z}_m$ along the directions of an orthonormal basis $\mathbf{U}_m$. As $\mathcal{M}$ is locally Euclidean, we compute the dimensions of the subspace $\mathbf{Z}_m$ by applying Gram-Schmidt (Dukes, 2014) to orthogonalize the span of representative local points. We formalize GBSM as follows:

$$\arg\min_{\delta_m, \theta'_m} \varrho(\delta_m, \theta'_m) := \sum_{z_m} \left\| z'_m - [z_m + \delta_m \odot \text{sign}(u_{im})] \right\|_2 \quad \text{where } z'_m \sim p(z'|x_i; \theta'_m).$$

The intuition behind GBSM is to utilize the fact that topological spaces are closed under their basis vectors to render $\mathcal{M}$ invariant to the perturbations $\delta_m$. To elaborate more on GBSM, we first sample a model instance $\theta'_m$. Then, we generate $z'_m \sim p(z'|x; \theta'_m)$ for all $x \in \mathbf{X}$. We orthogonalize $\mathbf{Z}_m$ and find the perturbations $\delta_m$ that minimizes $\varrho$ along the directions of the basis vectors $u_{im} \in \mathbf{U}_m$. We want the perturbations $\delta_m$ to be small. With $\delta_m$ fixed, we update $\theta'_m$ by minimizing $\varrho$ again. We use the notation GBSM$(\Theta', \Delta)$ where $\Delta = \{\delta_m\}_{m=1}^M$ to denote one update of $\Theta'$ via GBSM.

**Manifold Alignment.** Although GBSM confers us latent noise imperceptibility and sampling speed, $\Theta'$ may deviate from $\Theta$; in which case the manifolds they learn will mis-align. To mitigate this issue, we regularize each $\theta'_m \in \Theta'$ after every GBSM update. In essence, we apply one SVGD update on $\Theta'$ to ensure that $\Theta'$ follows the transform maps constructed by the particles $\Theta$ (Han & Liu, 2017).

$$\theta'_{t+1} \leftarrow \theta'_t + \alpha_t \pi(\theta'_t) \text{ where } \pi(\theta'_t) = \frac{1}{M} \sum_{m=1}^M \left[ k(\theta'_t, \theta^m_t) \nabla_{\theta^m_t} \log p(\theta^m_t) + \nabla_{\theta^m_t} k(\theta'_t, \theta^m_t) \right] \quad (4)$$

We use the notation SVGD$_\pi(\Theta')$ to refer to the gradient update rule in Equation 4. In this rule, the model instances $\Theta'$ determine their own update direction by consulting only the particles $\Theta$ instead of consulting each other. Maintaining $\Theta' = \{\theta'_m\}_{m=1}^M$ for large $M$ is, however, computationally prohibitive. Thus, as in Section 3, we keep only one (recognition) network $f_{\eta'}$ that takes as input $\xi'_m \sim \mathcal{N}(\mathbf{0}, \mathbf{I})$ and outputs $\theta'_m \sim f(\xi'_m; \eta')$. Here too we refine $\eta'$ through a small number of gradient steps to learn the trajectories that $\Theta'$ follows as it gets updated via GBSM and SVGD$_\pi$.

$$\eta'^{t+1} \leftarrow \arg\min_{\eta'} \sum_{m=1}^M \left\| \underbrace{f(\xi'_m; \eta'^t)}_{\theta'^t_m} - \theta'^{t+1}_m \right\|_2 \quad \text{where } \theta'^{t+1}_m \leftarrow \theta'^t_m + \alpha_t \pi(\theta'^t_m). \quad (5)$$

## 5 GENERATING ADVERSARIAL EXAMPLES

In this paper, a *black-box* scenario is considered. In this scenario, we have only access to the predictions of the classifier $g$. We produce adversarial examples by optimizing the loss below. The first term is the reconstruction loss inherent to VAEs. This loss accounts here for the dissimilarity between any input $x \in \mathcal{D}$ and its adversarial counterpart $x'$, and is constrained to be smaller than $\epsilon_{\text{attack}}$ so that $x'$ resides within an $\epsilon_{\text{attack}}$-radius ball of $x$. Unless otherwise specified, we shall use *norm $L_2$* as reconstruction loss. The second term is an auxiliary log-likelihood loss (for g) of a target class $y' \in \mathcal{Y} \setminus \{y\}$ where $y$ is the class of $x$. This loss defines the cost incurred for failing to fool $g$.

$$\mathcal{L}_{x'} = \|x - x'\|_2 + \min_{y' \in \mathcal{Y}} \left[ \mathbb{1}_{y=y'} \cdot \log\left(1 - P(y'|x')\right) \right] \text{ such that } \|x - x'\|_2 \le \epsilon_{\text{attack}}. \quad (6)$$

In Algorithm 2, we show how we unify our manifold learning and perturbation strategy into one learning procedure to generate adversarial examples without resorting to an exhaustive search.

## 6 RELATED WORK

**Manifold Learning.** VAEs are generally used to learn manifolds (Yu et al., 2018; Falorsi et al., 2018; Higgins et al., 2016) by maximizing the ELBO of the data log-likelihood (Alemi et al., 2017; Chen et al., 2017). Optimizing the ELBO entails reparameterizing the encoder to a Gaussian distribution (Kingma & Welling, 2014). This reparameterization is, however, restrictive (Jimenez Rezende & Mohamed, 2015) as it may lead to learning poorly the manifold of the data (Zhao et al., 2017). To alleviate this issue, we use SVGD, similar to Pu et al. (2017). While our approach and that of Pu et al. (2017) may look similar, ours is more principled. As discussed in (Hron et al., 2017), dropout which Pu et al. (2017) use is not Bayesian. Since our model instances are Bayesian, we are better equipped to capture the uncertainty. Capturing the uncertainty requires, however, evaluating the data likelihood. As we are operating in latent space, this raises the interesting challenge of assigning target-dependent variables to the inputs. We overcome this challenge using our inversion process.

**Adversarial Examples.** Studies in adversarial deep learning (Athalye et al., 2018; Kurakin et al., 2016; Goodfellow et al., 2014b; Athalye et al., 2017) can be categorized into two groups. The first group (Carlini & Wagner, 2016; Athalye et al., 2017; Moosavi-Dezfooli et al., 2016) proposes to

---

**Algorithm 2** *Generating Adversarial Examples*. Lines 2 and 4 compute distances between sets keeping a one-to-one mapping between them. $x'$ is adversarial to $x$ when $\mathcal{L}_{x'} \leq \epsilon_{\text{attack}}$ and $y \neq y'$.

1: **function** INNERTRAINING$(\boldsymbol{\Theta}, \boldsymbol{\Theta}', \boldsymbol{\eta}, \boldsymbol{\eta}', \boldsymbol{\Delta}, \tilde{x})$  ▷ local gradient updates of $f_\eta, f_{\eta'}, \Delta$
   **Require:** Learning rates $\boldsymbol{\beta}, \boldsymbol{\beta}'$
2:     $\boldsymbol{\eta} \leftarrow \boldsymbol{\eta} - \boldsymbol{\beta}\nabla_{\boldsymbol{\eta}}\|\boldsymbol{\Theta} - \text{SVGD}_\tau(\boldsymbol{\Theta})\|_2$  ▷ apply *inversion* on $\tilde{x}$ and update $\eta$
3:     $\boldsymbol{\Delta}, \boldsymbol{\Theta}' \leftarrow \text{GBSM}(\boldsymbol{\Theta}', \boldsymbol{\Delta})$  ▷ update $\Delta$ and $\Theta'$ using GBSM
4:     $\boldsymbol{\eta}' \leftarrow \boldsymbol{\eta}' - \boldsymbol{\beta}'\nabla_{\boldsymbol{\eta}'}\|\boldsymbol{\Theta}' - \text{SVGD}_\pi(\boldsymbol{\Theta}')\|_2$  ▷ align $\Theta'$ with $\Theta$ and update $\eta'$
5:     **return** $\boldsymbol{\eta}, \boldsymbol{\eta}', \boldsymbol{\Delta}$
**Require:** Training samples $(x, y) \in \boldsymbol{\mathcal{D}} \times \boldsymbol{\mathcal{Y}}$
**Require:** Number of model instances $M$
**Require:** Number of inner updates $T$
**Require:** Initialize weights $\boldsymbol{\eta}, \boldsymbol{\eta}', \boldsymbol{\phi}$  ▷ recognition nets $f_\eta, f_{\eta'}$, decoder $p_\phi$
**Require:** Initialize perturbations $\boldsymbol{\Delta} := \{\boldsymbol{\delta_m}\}_{m=1}^M$  ▷ latent (adversarial) perturbations
**Require:** Learning rates $\boldsymbol{\epsilon}, \boldsymbol{\alpha}, \boldsymbol{\alpha}'$, and noise margin $\epsilon_{\text{attack}}$
6: Sample $\boldsymbol{\xi_1}, ..., \boldsymbol{\xi_M}$ from $\mathcal{N}(\boldsymbol{0}, \mathbf{I})$  ▷ inputs to recognition nets $f_\eta, f_{\eta'}$
7: **for** $t = 1$ **to** $T$ **do**
8:     Sample $\boldsymbol{\Theta} = \{\boldsymbol{\theta_m}\}_{m=1}^M$ where $\boldsymbol{\theta_m} \sim \boldsymbol{f_\eta}(\boldsymbol{\xi_m})$
9:     Sample $\boldsymbol{\Theta}' = \{\boldsymbol{\theta'_m}\}_{m=1}^M$ where $\boldsymbol{\theta'_m} \sim \boldsymbol{f_{\eta'}}(\boldsymbol{\xi_m})$
10:    Use $\boldsymbol{\Theta}$ and $\boldsymbol{\Theta}'$ in Equation 3 to sample $z$ and $z'$
11:    Sample $\tilde{x} \sim p(x|z, \boldsymbol{\phi})$ and $x' \sim p(x'|z', \boldsymbol{\phi})$  ▷ clean and perturbed reconstructions
12:    $\boldsymbol{\eta}, \boldsymbol{\eta}', \boldsymbol{\Delta} \leftarrow \text{InnerTraining}(\boldsymbol{\Theta}, \boldsymbol{\Theta}', \boldsymbol{\eta}, \boldsymbol{\eta}', \boldsymbol{\Delta}, \tilde{x})$
13:    $\boldsymbol{\mathcal{L}_{\tilde{x}}} := \|x - \tilde{x}\|_2; \quad \boldsymbol{\mathcal{L}_{x'}} := \|x - x'\|_2$  ▷ reconstruction losses on $\tilde{x}$ and $x'$
14:    $\boldsymbol{\mathcal{L}_{x'}} := \begin{cases} \boldsymbol{\mathcal{L}_{x'}}, & \text{if } \mathcal{L}_{x'} > \epsilon_{\text{attack}} \\ \boldsymbol{\mathcal{L}_{x'}} + \min_{y' \in \mathcal{Y}} \left[ \mathbb{1}_{y=y'} \cdot \log(1 - P(y'|x')) \right], & \text{otherwise} \end{cases}$
15: $\boldsymbol{\eta} \leftarrow \boldsymbol{\eta} - \boldsymbol{\alpha}\nabla_{\boldsymbol{\eta}}\boldsymbol{\mathcal{L}_{\tilde{x}}}; \quad \boldsymbol{\eta}' \leftarrow \boldsymbol{\eta}' - \boldsymbol{\alpha}'\nabla_{\boldsymbol{\eta}'}\boldsymbol{\mathcal{L}_{x'}}$  ▷ SGD update using Adam optimizer
16: $\boldsymbol{\phi} \leftarrow \boldsymbol{\phi} - \boldsymbol{\epsilon}\nabla_{\boldsymbol{\phi}}(\boldsymbol{\mathcal{L}_{\tilde{x}}} + \boldsymbol{\mathcal{L}_{x'}})$  ▷ SGD update using Adam optimizer

---

generate adversarial examples directly in the input space of the original data by distorting, occluding or changing illumination in images to cause changes in classification. The second group (Song et al., 2018; Zhao et al., 2018b), where our work belongs, uses generative models to search for adversarial examples in the dense and continuous representations of the data rather than in its input space.

*Adversarial Images.* Song et al. (2018) propose to construct unrestricted adversarial examples in the image domain by training a conditional GAN that constrains the search region for a latent code $z'$ in the neighborhood of a target $z$. Zhao et al. (2018b) use also a GAN to map input images to a latent space where they conduct their search for adversarial examples. These studies are the closest to ours. Unlike in (Song et al., 2018) and (Zhao et al., 2018b), however, our adversarial perturbations are learned, and we do not constrain the search for adversarial examples to uniformly-bounded regions. In stark contrast to Song et al. (2018) and Zhao et al. (2018b) approaches also, where the search for adversarial examples is exhaustive and decoupled from the training of the GANs, our approach is efficient and end-to-end. Lastly, by capturing the uncertainty induced by embedding the data, we characterize the semantics of the data better, allowing us thus to generate sound adversarial examples.

*Adversarial Text.* Previous studies on adversarial text generation (Zhao et al., 2018a; Jia & Liang, 2017; Alvarez-Melis & Jaakkola, 2017; Li et al., 2016) perform word erasures and replacements directly in the input space using domain-specific rules or heuristics, or they require manual curation. Similar to us, Zhao et al. (2018b) propose to search for textual adversarial examples in the latent representation of the data. However, in addition to the differences aforementioned for images, the search for adversarial examples is handled more gracefully in our case thanks to an efficient gradient-based optimization method in lieu of a computationally expensive search in the latent space.

## 7 EXPERIMENTS & RESULTS

Before, we presented an attack model whereby we align the semantics of the inputs with their adversarial counterparts. As a reminder, our attack model is *black-box*, restricted and *non-targeted*. Our adversarial examples reside within an $\epsilon_{\text{attack}}$−radius ball of the inputs as our *reconstruction*

*loss, which measures the amount of changes in the inputs, is bounded by $\epsilon_{attack}$* (see Equation 6). We validate the adversarial examples we produce based on three evaluation criteria: (i.) *manifold preservation*, (ii.) *adversarial strength*, and (iii.) *soundness* via manual evaluation. We provide in Appendix A examples of the adversarial images and sentences that we construct.

## 7.1 MANIFOLD PRESERVATION

We experiment with a 3D non-linear Swiss Roll dataset which comprises 1600 datapoints grouped in 4 classes. We show in Figure 3, on the left, the 2D plots of the manifold we learn. In the middle, we plot the manifold and its elements that we perturbed and whose reconstructions are adversarial. On the right, we show the manifold overlaid with the latent codes of the adversarial examples produced by PGD (Goodfellow et al., 2014b) with $\epsilon_{attack} \leq 0.3$. Observe in Figure 3, in the middle, how the latent codes of our adversarial examples espouse the Swiss Roll manifold, unlike the plot on the right.

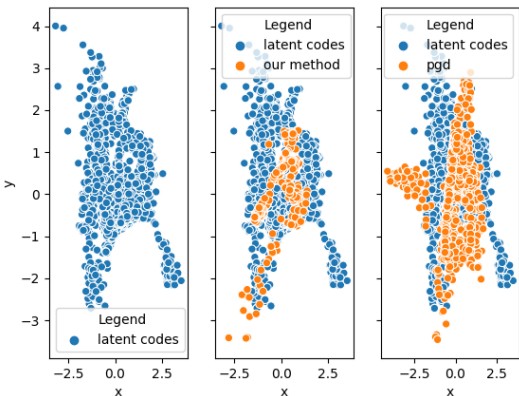

Figure 3: Invariance. Swiss Roll manifold learned with our encoder $E$ (left), and after perturbing its elements with our encoder $E'$ (middle) vs. that of PGD adversarial examples (right) learned using $E$.

## 7.2 ADVERSARIAL STRENGTH

**Setup.** As argued in (Athalye et al., 2018), the strongest non-certified defense against adversarial attacks is adversarial training with Projected Gradient Descent (PGD) (Goodfellow et al., 2014b). Thus, we evaluate the strength of our MNIST, CelebA and SVHN adversarial examples against adversarially trained ResNets (He et al., 2015) with a 40-step PGD and noise margin $\epsilon_{attack} \leq 0.3$. The ResNet models follow the architecture design of Song et al. (2018). Similar to Song et al. (2018) — whose attack model resembles ours[2] —, for MNIST, we also target the certified defenses (Raghunathan et al., 2018; Kolter & Wong, 2017) with $\epsilon_{attack} = 0.1$ using *norm $L_\infty$* as reconstruction loss. For all the datasets, the accuracies of the models we target are higher than 96.3%. Next, we present our attack success rates and give examples of our adversarial images in Figure 4.

**Attack Success Rate (ASR)** is the percentage of examples misclassified by the adversarially trained Resnet models. For $\epsilon_{attack} = 0.3$, the publicly known ASR of PGD attacks on MNIST is 88.79%. However, our ASR for MNIST is 97.2%, higher than PGD. Also, with $\epsilon_{attack} \approx 1.2$, using *norm $L_\infty$* as reconstruction loss in Equation 6, we achieve an ASR of 97.6% against (Kolter & Wong, 2017) where PGD achieves 91.6%. Finally, we achieve an ASR of 87.6% for SVHN, and 84.4% for CelebA.

### 7.2.1 ADVERSARIAL TEXT

**Datasets.** For text, we consider the SNLI (Bowman et al., 2015) dataset. SNLI consists of sentence pairs where each pair contains a premise ($P$) and a hypothesis ($H$), and a label indicating the relationship (*entailment, neutral, contradiction*) between the premise and hypothesis. For instance, the following pair is assigned the label *entailment* to indicate that the premise entails the hypothesis. *Premise: A soccer game with multiple males playing. Hypothesis: Some men are playing a sport.*

---

[2]Note that our results are, however, not directly comparable with (Song et al., 2018) as their reported success rates are for unrestricted adversarial examples, unlike ours, manually computed from Amazon MTurkers votes.

Table 1: Test samples and their perturbed versions. See more examples in Appendix A.

| | |
|---|---|
| True Input 1 | *P*: A group of people are gathered together. ***H*: There is a group here.** *Label*: Entailment |
| Adversary 1 | ***H'*: There is a group there.** *Label*: Contradiction |
| True Input 2 | *P*: A female lacrosse player jumps up. ***H*: A football player sleeps.** *Label*: Contradiction |
| Adversary 2 | ***H'*: A football player sits.** *Label*: Neutral |
| True Input 3 | *P*: A man stands in a curvy corridor. ***H'*: A man runs down a back alley.** *Label*: Contradiction |
| Adversary 3 | ***H'*: A man runs down a ladder alley.** *Label*: Neutral |

**Setup.** We perturb the hypotheses while keeping the premises unchanged. Similar to Zhao et al. (2018b), we generate adversarial text at word level using a vocabulary of 11,000 words. We also use ARAE (Zhao et al., 2018a) for word embedding, and a CNN for sentence embedding. To generate perturbed hypotheses, we experiment with three decoders: (i.) $p_\phi$ is a transpose CNN, (ii.) $p_\phi$ is a language model, and (iii.) we use the decoder of a pre-trained ARAE (Zhao et al., 2018a) model. We detail their configurations in Appendix B.

The transpose CNN generates more meaningful hypotheses than the language model and the pre-trained ARAE model although we notice sometimes changes in the meaning of the original hypotheses. We discuss these limitations more in details in Appendix A. Henceforward, we use the transpose CNN to generate perturbed hypotheses. See Table 1 for examples of generated hypotheses.

**Attack Success Rate (ASR).** We attack an SNLI classifier that has a test accuracy of 89.42%. Given a pair (*P*, *H*) with label *l*, its perturbed version (*P*, *H'*) is adversarial if the classifier assigns the label *l* to (*P*, *H*), (*P*, *H'*) is manually found to retain the label of (*P*, *H*), and such label differs from the one the classifier assigns to (*P*, *H'*). To compute the ASR, we run a pilot study which we detail next.

### 7.3 MANUAL EVALUATION

To validate our adversarial examples and assess their soundness vs. Song et al. (2018), Zhao et al. (2018b) and PGD (Madry et al., 2017) adversarial examples, we carry out a pilot study whereby we ask three yes-or-no questions: (Q1) *are the adversarial examples semantically sound?*, (Q2) *are the true inputs similar perceptually or in meaning to their adversarial counterparts?* and (Q3) *are there any interpretable visual cues in the adversarial images that support their misclassification?*

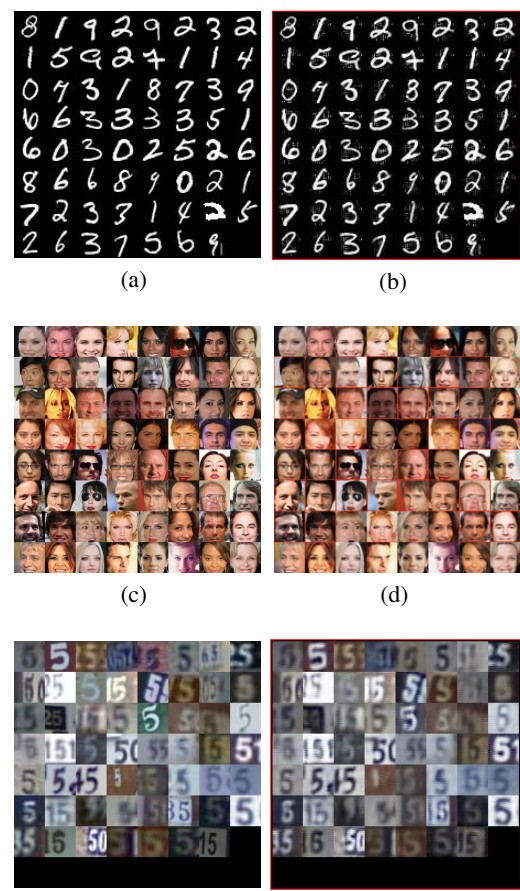

(a)      (b)

(c)      (d)

(e)      (f)

Figure 4: Inputs (left) - Adversarial examples (right, inside red boxes). MNIST: (a)-(b), CelebA: (c)-(d), SVHN: (e)-(f). See **Appendix A** for more samples with higher resolution.

**Pilot Study I.** For MNIST, we pick 50 images (5 for each digit), generate their clean reconstructions, and their adversarial examples against a 40-step PGD ResNet with $\epsilon_{\text{attack}} \leq 0.3$. We target also the certified defenses of Raghunathan et al. (2018) and Kolter & Wong (2017) with $\epsilon_{\text{attack}} = 0.1$. For SVHN, we carry out a similar pilot study and attack a 40-step PGD ResNet. For CelebA, we pick 50 images (25 for each gender), generate adversarial examples against a 40-step PGD ResNet. For all three datasets, we hand the images and the questionnaire to 10 human subjects for manual evaluation. We report in Table 2 the results for MNIST and, in Table 4, the results for CelebA and SVHN.

We hand the same questionnaire to the subjects with 50 MNIST images, their clean reconstructions, and the adversarial examples we craft with our method. We also handed the adversarial examples

generated using Song et al. (2018), Zhao et al. (2018b) and PGD methods. We ask the subjects to assess the soundness of the adversarial examples based on the *semantic features* (e.g., *shape, distortion, contours, class*) of the real MNIST images. We report the evaluation results in Table 3.

Table 2: Pilot Study I. Note that against the certified defenses of Raghunathan et al. (2018) and Kolter & Wong (2017), Song et al. (2018) achieved (manual) success rates of 86.6% and 88.6%.

| QUESTIONNAIRE | MNIST | | |
|---|---|---|---|
| | 40-STEP PGD | RAGHUNATHAN ET AL. (2018) | KOLTER & WONG (2017) |
| QUESTION Q1: YES | 100 % | 100 % | 100 % |
| QUESTION Q2: YES | 100 % | 100 % | 100 % |
| QUESTION Q3: NO | 100 % | 100 % | 100 % |

Table 3: Pilot Study I. The adversarial images are generated against the adversarially trained Resnets.

| QUESTIONNAIRE | OUR METHOD | SONG ET AL. (2018) | ZHAO ET AL. (2018B) | PGD |
|---|---|---|---|---|
| QUESTION Q1: YES | 100 % | 85.9 % | 97.8 % | 76.7 % |
| QUESTION Q2: YES | 100 % | 79.3 % | 89.7 % | 66.8 % |
| QUESTION Q3: NO | 100 % | 71.8 % | 94.6 % | 42.7 % |

**Pilot Study II - SNLI.** Using the transpose CNN as decoder $p_\phi$, we generate adversarial hypotheses for the SNLI sentence pairs with the premises kept unchanged. Then, we select manually 20 pairs of clean sentences (premise, hypothesis), and adversarial hypotheses. We also pick 20 pairs of sentences and adversarial hypotheses generated this time using Zhao et al. (2018b)'s method against their treeLSTM classifier. We choose this classifier as its accuracy (89.04%) is close to ours (89.42%). We carry out a pilot study where we ask two yes-or-no questions: (Q1) *are the adversarial samples semantically sound?* and (Q2) *are they similar to the true inputs?* We report the results in Table 4.

Table 4: Pilot Studies. [†] Some adversarial images and original ones were found blurry to evaluate.

| QUESTIONNAIRE | PILOT I | | PILOT II - SNLI | |
|---|---|---|---|---|
| | CELEBA | SVHN | OUR METHOD | ZHAO ET AL. (2018B) |
| QUESTION Q1: YES | 100 % | 95[†] % | 83.7 % | 79.6% |
| QUESTION Q2: YES | 100 % | 97 % | 60.4 % | 56.3% |
| QUESTION Q3: NO | 100 % | 100 % | N/A | N/A |

**Takeaways.** As reflected in the pilot study and the attack success rates, we achieve good results in the image and text classification tasks. In the image classification task, we achieve better results than PGD and Song et al. (2018) both against the certified and non-certified defenses. The other takeaway is: although the targeted certified defenses are resilient to adversarial examples crafted in the input space, we can achieve manual success rates higher than the certified rates when the examples are constructed in the latent space, and the search region is unrestricted. In text classification, we achieve better results than Zhao et al. (2018b) when using their treeLSTM classifier as target model.

## 8 CONCLUSION

Many approaches in adversarial attacks fail to enforce the semantic relatedness that ought to exist between original inputs and their adversarial counterparts. Motivated by this fact, we developed a method tailored to ensuring that the original inputs and their adversarial examples exhibit similar semantics by conducting the search for adversarial examples in the manifold of the inputs. Our success rates against certified and non-certified defenses known to be resilient to traditional adversarial attacks illustrate the effectiveness of our method in generating sound and strong adversarial examples.

Although in the text classification task we achieved good results and generated informative adversarial sentences, as the transpose CNN gets more accurate — *recall that it is partly trained to minimize a*

*reconstruction error* —, generating adversarial sentences that are different from the input sentences and yet preserve their semantic meaning becomes more challenging. In the future, we intend to build upon the recent advances in text understanding to improve our text generation process.

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

## APPENDIX A: DISCUSSION & ADVERSARIAL EXAMPLES

**Posterior Formulation.** Similar to (Kim et al., 2018), we formalize $p(\theta|\mathcal{D})$ for every $\theta \in \Theta$ as:

$$p(\theta|\mathcal{D}) \propto p(\mathcal{D}|\theta)p(\theta) = \prod_{(x,\tilde{z})} p(\tilde{z}|x;\theta)p(\theta) \text{ where } x \in \mathcal{D} \text{ and } \tilde{z} \text{ is generated using Algorithm 1}$$

$$= \prod_{(x,\tilde{z})} \mathcal{N}(\tilde{z}|f_W(x),\gamma^{-1})\mathcal{N}(W|f_\eta(\xi),\lambda^{-1})\text{Gamma}(\gamma|a,b)\text{Gamma}(\lambda|a',b')$$

For every $\theta' \in \Theta'$, we compute $p(\theta'|\mathcal{D})$ the same way. Note that $\theta$ (resp. $\theta'$) consists in fact of network parameters $W \sim f_\eta$ (resp. $W' \sim f_{\eta'}$) and scaling parameters $\gamma$ and $\lambda$. For notational simplicity, we used before the shorthands $\theta \sim f_\eta$ and $\theta' \sim f_{\eta'}$. The parameters $\gamma$ and $\lambda$ are initially sampled from a Gamma distribution and updated as part of the learning process. In our experiments, we set the hyper-parameters of the Gamma distributions $a$ and $b$ to 1.0 and 0.1, and $a'$ and $b'$ to 1.0.

**Latent Noise Level.** We measure the amount of noise $\Delta$ we inject into the latent codes of our inputs by computing the average spectral norm of the latent codes of their adversarial counterparts. The input changes are captured by our reconstruction loss which is bounded by $\epsilon_{\text{attack}}$ (see Equation 6). For MNIST, CelebA, and SVHN, the noise levels are $0.004 \pm 0.0003, 0.026 \pm 0.005$, and $0.033 \pm 0.008$. The takeaways are: (i.) they are imperceptible, and (ii.) the distributions that $\Theta$ and $\Theta'$ follow are similar. To validate (ii.), we compute the marginals of clean and perturbed latent codes randomly sampled from $\Theta$ and $\Theta'$. As shown in Figure 5, the marginal distributions overlap relatively well.

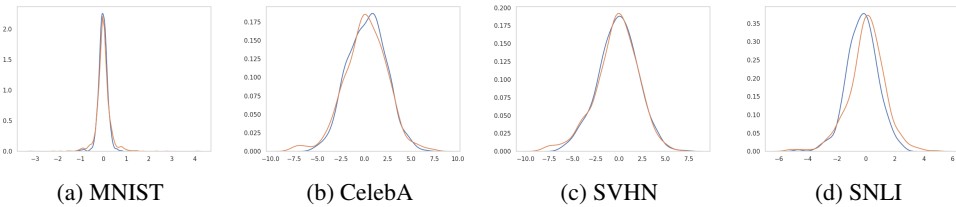

| (a) MNIST | (b) CelebA | (c) SVHN | (d) SNLI |

Figure 5: Marginal distributions of clean (blue) and perturbed (red) latent codes over few minibatches.

**Discussion.** We discuss the choices pertaining to the design of our approach and their limitations. We discuss also the evaluation process of our approach against (Song et al., 2018; Zhao et al., 2018b).

*Space/Time Complexity.* As noted in (Jimenez Rezende & Mohamed, 2015), the Gaussian prior assumption in VAEs might be too restrictive to generate meaningful enough latent codes (Zhao et al., 2017). To relax this assumption and produce informative and diverse latent codes, we used SVGD. To generate manifold preserving adversarial examples, we proposed GBSM. Both SVGD and GBSM maintain a set of $M$ model instances. As ensemble methods, both inherit the shortcomings of ensemble models most notably in space/time complexity. Thus, instead of maintaining $2 * M$ model instances, we maintain only $f_\eta$ and $f_{\eta'}$ from which we sample these model instances. We experimented with $M$ set to $2, 5, 10$ and 15. As $M$ increases, we notice some increase in sample quality at the expense of longer runtimes. The overhead that occurs as $M$ takes on larger values reduces, however, drastically during inference as we need only $f_{\eta'}$ to sample the model instances $\theta'_m \in \Theta'$ in order to construct adversarial examples. One way to alleviate the overhead during training is to enforce weight-sharing for $\theta_m \in \Theta$ and $\theta'_m \in \Theta'$. However, we did not try this out.

*Preserving Textual Meaning.* To construct adversarial text, we experimented with three architecture designs for the decoder $p_\phi$: (i.) a transpose CNN, (ii.) a language model, and (iii.) the decoder of a pre-trained ARAE model (Zhao et al., 2018a). The transpose CNN generates more legible text than the other two designs although we notice sometimes some changes in meaning in the generated adversarial examples. Adversarial text generation is challenging in that small perturbations in the latent codes can go unnoticed at generation whereas high noise levels can render the outputs nonsensical. To produce adversarial sentences that faithfully preserve the meaning of the inputs, we need good sentence generators, like GPT (Radford, 2018), trained on large corpora. Training such large language models requires however time and resources. Furthermore, in our experiments, we considered only a vocabulary of size 10,000 words and sentences of length no more than 10 words to align our evaluation with the experimental choices of (Zhao et al., 2018b).

*Measuring Perceptual Quality* is desirable when the method relied upon to generate adversarial examples uses GANs or VAEs; both known to produce often samples of limited quality. As Song et al. (2018) perform unrestricted targeted attacks — *their adversarial examples might totally differ from the true inputs* — and Zhao et al. (2018b) do not target certified defenses, a fair side-by-side comparison of our results and theirs using metrics like *mutual information* or *frechet inception distance*, seems unachievable. Thus, to measure the quality of our adversarial examples and compare our results with (Song et al., 2018) and (Zhao et al., 2018b), we carried out the pilot study.

ADVERSARIAL IMAGES: CELEBA

Table 5: CelebA samples, their clean reconstructions, and adversarial examples (in red boxes).

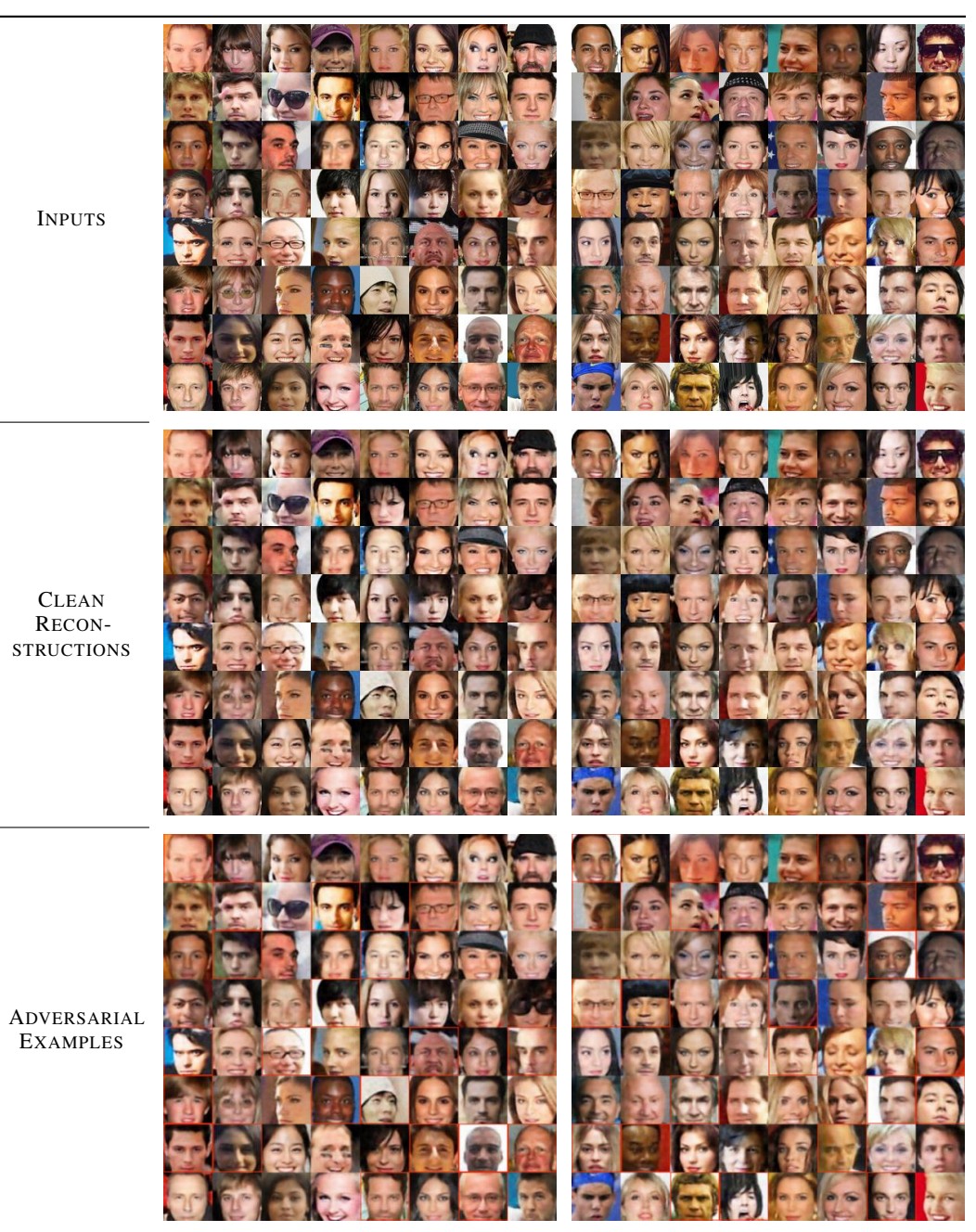

ADVERSARIAL IMAGES: SVHN

Here, we provide few random samples of non-targeted adversarial examples we generate with our approach on the SVHN dataset as well as the clean reconstructions.

Table 6: SVHN. Images in red boxes are all adversarial.

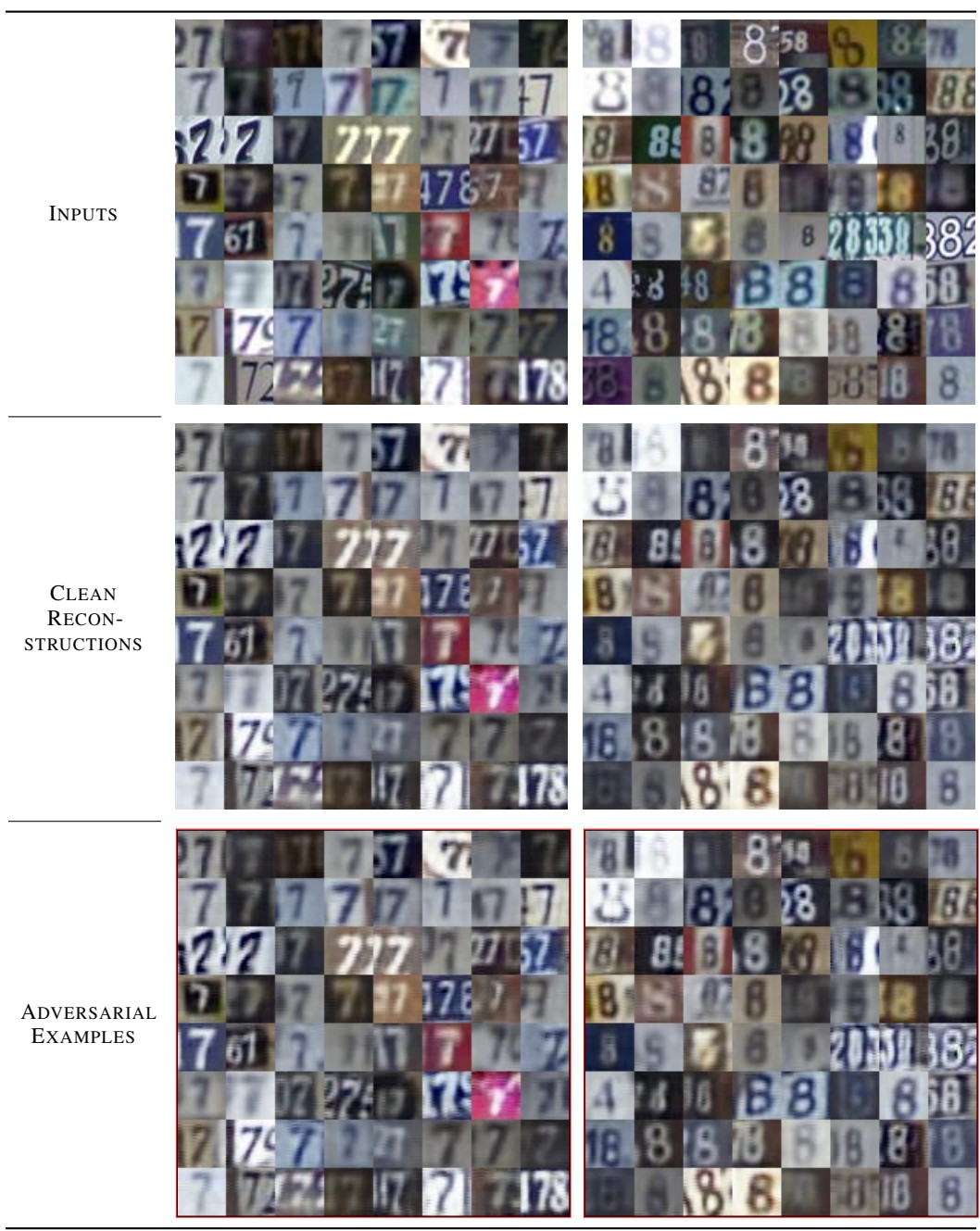

ADVERSARIAL IMAGES: MNIST

Here, we provide few random samples of non-targeted adversarial examples we generate with our approach on the MNIST dataset as well as the clean reconstructions. Both the reconstructed and the adversarial images look realistic although we notice some artifacts on the latter. Basic Iterative Methods (Kurakin et al., 2016), among others, also suffer from this. Because the marginal distributions of the latent codes of the inputs and their perturbed versions overlap, we conclude that indeed the adversarial images preserve the manifold of the inputs (see Figure 5).

Table 7: MNIST. Images in red boxes are all adversarial.

ADVERSARIAL TEXT: SNLI

Table 8: Examples of adversarially generated hypotheses with the true premises kept unchanged.

| | |
|---|---|
| TRUE INPUT 1 | *P*: A white dog is running through the snow.
*H*: **A CAT STALKING THROUGH THE SNOW.**
*Label*: CONTRADICTION |
| ADVERSARY | *H'*: **A CAT HOPS IN THE SNOW.** *Label*: NEUTRAL |
| TRUE INPUT 2 | *P*: Three dogs are searching for something outside.
*H*: **THERE ARE FOUR DOGS.**
*Label*: CONTRADICTION |
| ADVERSARY | *H'*: **THERE ARE FIVE DOGS.** *Label*: NEUTRAL |
| TRUE INPUT 3 | *P*: A man waterskis while attached to a parachute.
*H*: **A BULLDOZER KNOCKS DOWN A HOUSE.**
*Label*: CONTRADICTION |
| ADVERSARY | *H'*: **A BULLDOZER KNOCKS DOWN A CAGE.** *Label*: ENTAILMENT |
| TRUE INPUT 4 | *P*: A little girl playing with flowers.
*H*: **A LITTLE GIRL PLAYING WITH A BALL.**
*Label*: CONTRADICTION |
| ADVERSARY | *H'*: **A LITTLE GIRL IS RUNNING WITH A BALL.** *Label*: NEUTRAL |
| TRUE INPUT 5 | *P*: People stand in front of a chalkboard.
*H*: **PEOPLE STAND OUTSIDE A PHOTOGRAPHY STORE.**
*Label*: CONTRADICTION |
| ADVERSARY | *H'*: **PEOPLE STAND IN FRONT OF A WORKSHOP.** *Label*: NEUTRAL |
| TRUE INPUT 6 | *P*: Musician entertaining **his** audience.
*H*: **THE WOMAN PLAYED THE TRUMPET.**
*Label*: CONTRADICTION |
| ADVERSARY | *H'*: **THE WOMAN PLAYED THE DRUMS.** *Label*: ENTAILMENT |
| TRUE INPUT 7 | *P*: A kid on a slip and slide.
*H*: **A SMALL CHILD IS INSIDE EATING THEIR DINNER.**
*Label*: CONTRADICTION |
| ADVERSARY | *H'*: **A SMALL CHILD IS EATING THEIR DINNER.** *Label*: ENTAILMENT |
| TRUE INPUT 8 | *P*: A deer jumping over a fence.
*H*: **A DEER LAYING IN THE GRASS.** *Label*: CONTRADICTION |
| ADVERSARY | *H'*: **A PONY LAYING IN THE GRASS.**
*Label*: ENTAILMENT |
| TRUE INPUT 9 | *P*: Two vendors are on a curb selling balloons.
*H*: **THREE PEOPLE SELL LEMONADE BY THE ROAD SIDE.**
*Label*: CONTRADICTION |
| ADVERSARY | *H'*: **THREE PEOPLE SELL ARTWORK BY THE ROAD SIDE.**
*Label*: ENTAILMENT |

Table 9: Some generated examples deemed adversarial by our method that are not.

| | |
|---|---|
| TRUE INPUT 1 | *P*: A man is operating some type of a vessel.
*H*: **A DOG IN KENNEL.**
*Label*: CONTRADICTION |
| GENERATED | *H'*: **A DOG IN DISGUISE.** *Label*: CONTRADICTION |
| TRUE INPUT 2 | *P*: A skier.
*H*: **SOMEONE IS SKIING.**
*Label*: ENTAILMENT |
| GENERATED | *H'*: **MAN IS SKIING.** *Label*: NEUTRAL |
| TRUE INPUT 3 | *P*: This is a bustling city street.
*H*: **THERE ARE A LOT OF PEOPLE WALKING ALONG.**
*Label*: ENTAILMENT |
| GENERATED | *H'*: **THERE ARE A LOT GIRLS WALKING ALONG.** *Label*: NEUTRAL |
| TRUE INPUT 4 | *P*: A soldier is looking out of a window.
*H*: **THE PRISONER'S CELL IS WINDOWLESS.**
*Label*: CONTRADICTION |
| GENERATED | *H'*: **THE PRISONER'S HOME IS WINDOWLESS.** *Label*: CONTRADICTION |
| TRUE INPUT 5 | *P*: Four people sitting on a low cement ledge.
*H*: **THERE ARE FOUR PEOPLE.**
*Label*: ENTAILMENT |
| GENERATED | *H'*: **THERE ARE SEVERAL PEOPLE.** *Label*: NEUTRAL |
| TRUE INPUT 6 | *P*: Three youngsters shovel a huge pile of snow.
*H*: **CHILDREN WORKING TO CLEAR SNOW.**
*Label*: ENTAILMENT |
| GENERATED | *H'*: **KIDS WORKING TO CLEAR SNOW.** *Label*: NEUTRAL |
| TRUE INPUT 7 | *P*: Boys at an amphitheater.
*H*: **BOYS AT A SHOW.**
*Label*: ENTAILMENT |
| GENERATED | *H'*: **BOYS IN A SHOW.** *Label*: NEUTRAL |
| TRUE INPUT 8 | *P*: Male child holding a yellow balloon.
*H*: **BOY HOLDING BIG BALLOON.** *Label*: NEUTRAL |
| GENERATED | *H'*: **BOY HOLDING LARGE BALLOON.**
*Label*: NEUTRAL |
| TRUE INPUT 9 | *P*: Women in their swimsuits sunbathe on the sand.
*H*: **WOMEN UNDER THE SUN ON THE SAND.**
*Label*: ENTAILMENT |
| GENERATED | *H'*: **FAMILY UNDER THE SUN ON THE SAND.**
*Label*: NEUTRAL |

## APPENDIX B: EXPERIMENTAL SETTINGS

Table 10: Model Configurations + SNLI Classifier + Hyper-parameters.

|  | NAME | CONFIGURATION |
|---|---|---|
| RECOGNITION NETWORKS | $f_\eta$ | INPUT DIM: 50,
HIDDEN LAYERS: [60, 70],
OUTPUT DIM: NUM WEIGHTS & BIASES IN $\theta_m$ |
|  | $f'_\eta$ | INPUT DIM: 50,
HIDDEN LAYERS: [60, 70],
OUTPUT DIM: NUM WEIGHTS & BIASES IN $\theta'_m$ |
| MODEL INSTANCES | PARTICLES $\theta_m$ | INPUT DIM: $28 \times 28$ (MNIST),
$64 \times 64$ (CELEBA),
$32 \times 32$ (SVHN), 300 (SNLI)
HIDDEN LAYERS: [40, 40]
OUTPUT DIM (LATENT CODE): 100 |
|  | PARAMETERS $\theta'_m$ | INPUT DIM: $28 \times 28$ (MNIST),
$64 \times 64$ (CELEBA),
$32 \times 32$ (SVHN), 100 (SNLI)
HIDDEN LAYERS: [40, 40]
OUTPUT DIM (LATENT CODE): 100 |
| FEATURE EXTRACTOR |  | INPUT DIM: $28 \times 28 \times 1$ (MNIST), $64 \times 64 \times 3$ (CELEBA),
$32 \times 32 \times 3$ (SVHN), $10 \times 100$ (SNLI)
HIDDEN LAYERS: [40, 40]
OUTPUT DIM: $28 \times 28$ (MNIST), $64 \times 64$ (CELEBA),
$32 \times 32$ (SVHN), 100 (SNLI) |
| DECODER | TRANSPOSE CNN | FOR CELEBA & SVHN: [FILTERS: 64, STRIDE: 2,
KERNEL: 5]$\times$ 3
FOR SNLI: [FILTERS: 64, STRIDE: 1,
KERNEL: 5]$\times$ 3 |
|  | LANGUAGE MODEL | VOCABULARY SIZE: 11,000 WORDS
MAX SENTENCE LENGTH: 10 WORDS |
| SNLI CLASSIFIER |  | INPUT DIM: 200, HIDDEN LAYERS: [100, 100, 100], OUTPUT DIM: 3 |
| LEARNING RATES |  | ADAM OPTIMIZER $(\delta = 5.10^{-4}), \alpha = 10^{-3}, \beta = \beta' = 10^{-2}$ |
| MORE SETTINGS |  | BATCH SIZE: 64, INNER-UPDATES: 3, TRAINING EPOCHS: 1000, $M = 5$ |

