# OpenReview forum: "Semantics Preserving Adversarial Attacks"
_ICLR.cc/2020/Conference — Reject_

### Official Review · AnonReviewer1 · 2019-10-16
**Official Blind Review #1**

**Rating:** 6

**Review:**

Major caveat: I have published in the area of adversarial attacks on NLP models, but the specifics of the methods presented in this paper are quite outside of my expertise, and I do not have time to become familiar with them for this review.  I hope there are other reviewers that are more qualified than I am to check the specifics of the methods.

This paper presents a new technique for generating adversarial examples, by first learning the data manifold in an embedding space, then finding an adversarial example that lies on the manifold.  I like this idea, it intuitively seems like a promising method for obtaining semantically meaningful adversarial examples.

As I said above, I do not feel qualified to review whether the method should _theoretically_ accomplish its goals, so my judgment of this paper is on the intuition behind the idea (which I like), and the results that I can see (which are less promising).  In order to have a "semantics preserving" attack, the method needs to (1) remain on the data manifold, and (2) not change the label a human would give to the input.

For (1), this appears to have been accomplished on most datasets, though it seems pretty hard to argue that the artifacts seen in the MNIST examples shown are on the data manifold - there are no such artifacts in any of the inputs, or in the clean reconstruction.  How do the authors claim that this actually did a reasonable job of staying in the data manifold?

For (2), most of the images do indeed look like they should retain their human labels, which is good (but also not hard for adversarial images).  Almost all of the textual examples, however, have correct predictions from the model after the adversarial change to the input.  You can't really argue that these are "semantics preserving", or even "successful attacks", as they change the expected input label.  This is why semantics-preserving attacks are so hard in NLP, and I don't think that this method has accomplished its goal here at all, at least for text.  The authors should consult with experts in NLP before making claims about successfully constructing semantics preserving attacks on NLP models.

I'm pretty on the fence about this paper, as I like the intuition, and the method appears to work reasonably well for vision.  It does not work as claimed for text, however, and that should be fixed before this paper is published (either with softened claims or with better results).  Hopefully people from other perspectives can pipe in and give a more clear picture on this paper.

EDIT: See discussion below for my justification for reducing my score from a 3 to a 1.

EDIT 11/14: The authors' revisions have satisfied my concerns about how the NLP attacks are described.  I'm a little bit nervous about how the examples were changed - it seems that nothing changed about the method itself, so the authors probably cherry-picked better examples - but that's not sufficiently worrying to me to justify rejection.  The pilot study is also quite weak, as the number of inputs that were evaluated was only 20, and the questions presented don't appear to ask about changes in the label.  I don't know how you could get 100% on that given the examples that I saw in the previous version of the paper.  This is all to say that I don't think the NLP attacks are actively problematic anymore, as they were previously, now they are just weak.  The main contribution here is the technical contribution, anyway, so weaker results on one of the datasets tested is not a deal-breaker to me.  Assuming the technical contributions pass muster (which, as I said, I don't really feel qualified to judge), I'm satisfied with this paper as it is now.

**Experience Assessment:**

I have published one or two papers in this area.

**Review Assessment: Checking Correctness Of Derivations And Theory:**

I did not assess the derivations or theory.

**Review Assessment: Checking Correctness Of Experiments:**

I carefully checked the experiments.

**Review Assessment: Thoroughness In Paper Reading:**

I read the paper at least twice and used my best judgement in assessing the paper.

---

### Official Review · AnonReviewer2 · 2019-10-21
**Official Blind Review #2**

**Rating:** 6

**Review:**

The paper presents an approach to generating adversarial examples that preserve the semantics of the input examples. To do so, the approach reconstructs the manifold where the input examples lie and then generates new examples by perturbing the elements of the manifold so as to ensure the new elements remain in the manifold to preserve the semantics of the elements.

In the presented system the manifold is learned by means of Stein Variational Gradient Descent, while the perturbation is made by applying the Gram-Schmidt process which ensures that the perturbed elements still reside in the manifold.

To generate adversarial examples the approach presented in the paper considers a scenario in which only the predictions of the classifier are known, to be able to compute and optimize the loss function.

The presented approach has been tested on toy examples regarding images (both numbers from MNIST or SVHN and images from CelebA datasets) and texts (SNLI dataset).

The performance presented in the paper is promising. The results show that the manifold shape is preserved while creating perturbed elements. The system also achieves good results in terms of adversarial success rate, which however I believe should be called "attack success rate", a term widely used in literature.

The paper is well written and seems to me to be mathematically sound. I have very few comments on the paper which does not present any important lack in my opinion.

I suggest moving algorithm 2 to a new page in order to have the whole pseudo-code together.
Moreover, I have found two typos, one on page 4 in the last equation where z'_m I think it is wrongly written as z^'_m (the ' is far from the z), and one on page 6 in the "adversarial examples" paragraph where the word "data" is misspelt as "dat".

**Experience Assessment:**

I have read many papers in this area.

**Review Assessment: Checking Correctness Of Derivations And Theory:**

N/A

**Review Assessment: Checking Correctness Of Experiments:**

I assessed the sensibility of the experiments.

**Review Assessment: Thoroughness In Paper Reading:**

I made a quick assessment of this paper.

---

> ### Author Response · Authors · 2019-11-06
> **fixing typos and putting the algorithm in one page**
>
> We thank the reviewer for the useful comments and suggestions. We'll fix the typos and put the algorithm in one page. We'll also use the terminology "attack success rate" instead of "adversarial success in our updated paper.
>
> Given that these are the only issues that the reviewer raised, we believe we deserve, in all fairness, a higher rating. We stand ready though to address any other issues the reviewer might raise.

---

### Official Review · AnonReviewer3 · 2019-10-23
**Official Blind Review #3**

**Rating:** 1

**Review:**


====== Updates ======
I appreciate the authors' time and effort in the response. I have read the rebuttal, but I am not convinced by the authors' argument on using L2 (or L_\infty) constraints. No matter whether L2 or L_\infty constraint is used, the authors' method is not directly comparable to methods in Song et al. (2018), making the results in Table 2 and Table 3 meaningless and confusing.

- Song et al. (2018) indeed constraints the search region of latent code to be within a small L2 ball of a randomly sampled anchor latent code. However, this anchor latent code is not directly related to any given image in the dataset, and therefore the generated adversarial examples are not close to any existing image. In contrast, the authors' attack is still basically a norm-bounded attack, which is not directly comparable to the unrestricted attack in Song et al. (2018).

- Song et al. (2018) is a white box attack, while the attack in this paper is black box.

====== Original review =======

This paper proposes to generate semantic preserving adversarial examples by first learning a manifold and then perturbing data along the manifold. In this way the generated adversarial examples can be semantically close to the original clean examples, and the perturbations can be hopefully more natural. For manifold learning, the authors propose to use a similar approach to that proposed in Pu et al. (2017), which uses SVGD to train a VAE. After the VAE is trained, the authors use GBSM to train a model to produce semantic adversarial examples efficiently.

I have many concerns for this paper:

- The approach is not well motivated. It is unclear why using a fully Bayesian framework and employing SVGD to learn the VAE model is preferred for conducting semantic adversarial attacks. Many choices in the algorithm seem to be arbitrary, and there are many approximations in the method whose accuracies have no guarantees. For example, the recognition networks  are used to approximate the updated parameters of the encoder from SVGD. Sampling from the posterior distribution of z is approximated by first doing Monte Carlo over \Theta. For "manifold alignment" another recognition network is used to approximate the updates from SVGD. It is hard to predict how those approximation errors accumulate when all pieces are combined together to form a very complicated algorithm.

- In Equation (6) the authors hard-constrain the generated adversarial example such that they cannot differ from the original data by some pre-specified l_2-norm. This leads to many unfair comparisons in the experiments:

    1. The authors compare their approach to other attacking methods on the success rates of attacking Madry's model and Kolter & Wong's certified model. However, both Madry and Kolter & Wong's model are for attacks using the l_infinity norm. It is unfair that the authors' attack uses l_2 norm. In fact, it is known that models robust to l_infinity norm attacks are generally not robust to attacks using other norms.

    2. The authors also compare their approach to methods in Song et al. (2018) and Zhao et al. (2018a). However, the two previous approaches did not directly constrain the distance between generated adversarial examples and the corresponding clean inputs. Therefore, when using human evaluation to assess the image quality of generated adversarial examples, the two previous methods are naturally at a huge disadvantage. In stark contrast, the authors' adversarial images are constrained to be close to the corresponding unperturbed images under a small l_2 norm, which naturally have higher image quality.

**Experience Assessment:**

I have published one or two papers in this area.

**Review Assessment: Checking Correctness Of Derivations And Theory:**

N/A

**Review Assessment: Checking Correctness Of Experiments:**

I assessed the sensibility of the experiments.

**Review Assessment: Thoroughness In Paper Reading:**

I made a quick assessment of this paper.

---

> ### Author Response · Authors · 2019-11-06
> **On the need of using Stein and capturing the uncertainty**
>
> We thank the reviewer for the constructive comments.
>
> First, we believe we have motivated well enough the design choices of our approach. To clarify our choices even more, we want to generate semantics preserving adversarial examples. In order to achieve this goal, we need to characterize as faithfully as possible the semantics of our inputs. In that regard, we learn the manifold that captures the semantics of the inputs using VAEs. We introduce a variational inference method for manifold learning.
>
> Typically, VAEs learn an encoding function that maps the data manifold to an isotropic Gaussian, and a decoding function that transforms a latent code back to a sample in the input space. The reason for choosing the isotropic Gaussian is to render the ELBO tractable and hence easy to optimize. However, such a constraint imposed on the data manifold is quite restrictive and could lead to learning poorly the semantics of the data, as discussed in Jimenez Rezende & Mohamed (2015). Given that we want to generate semantics preserving adversarial examples, it is paramount that we characterize the semantics of the inputs well, while restraining ourselves from imposing rigid assumptions on the distribution of the data.
>
> In that regard, we proposed to optimize the second KL (see the paper), similar to Pu et al. (2017), using SVGD. With SVGD, we don't need to impose a parametric/functional form for the distribution we want to learn. Furthermore, SVGD combines the merits of MCMC methods, which represent a category of formal Bayesian approaches for parameter estimation and uncertainty quantification, and variational inference. Hence, it is just fitting to propose a Bayesian framework.
>
> We are *not* using another recognition network for "manifold alignment". The recognition network $f_{\eta'}$ generates $\Theta'$, and alleviates the burden of maintaining $M$ model instances. Pu et al. (2017) also used a similar approach. The "manifold alignment" is just a way to regularize the encoder E' that perturbs the latent codes generated by the encoder E.
>
> We also believe that it is a wide practice to use empirical risk minimization to approximate distributions. With respect to the errors accumulating due to the approximations we introduce, what matters is whether or not the distribution that $\Theta'$ follows is similar to the distribution that $\Theta$ follows as $\Theta$ parameterizes the encoder that learns the data manifold. To support our claims that both distributions are similar, we showed in the Appendix in Fig 5 that these distributions overlap quite well for most of the datasets we considered. Also, using toy data, we showed that the manifolds of the benign examples and the adversarial ones overlap quite well. Finally, the quality of the adversarial images is quite high compared to the baselines we considered.
>
> Regarding the use of the $L_2$ as a reconstruction loss, we are now running new experiments using $L_\infty$ loss and we will update the paper with our results.
>
> Indeed, Song et al. (2018) and Zhao et al. (2018b), did not constrain the distance between generated adversarial and corresponding clean inputs. However, they constrained their latent codes to be close. Also in both papers, they used GANs which learn to approximate the true data distribution. In that regard, we would expect the adversarial examples they generate to be as natural, as realistically looking, as the clean inputs, even more than with our approach where we just constrain the distance between the examples. However, this is not the case.
> We would be happy to discuss this more in the paper if the reviewer deems this worthy. We appreciate the reviewer's take on our comments.

---

### Author Response · Authors · 2019-11-14
**Rebuttals & Updates in the manuscript**

Reviewer 1: Thanks for the useful and constructive feedback. Your insights have helped us greatly improve the quality of our paper. We have changed the manual evaluation process of our generated text as well as how we evaluated the adversarial text generated using Zhao et al. (2018b)'s approach. Before, we considered any attack to be successful therefore the targeted classifier assigns a label to a perturbed sample that is different from the label it assigns to the clean input. Although this definition holds for images, as the reviewer pointed out, however, it is weak for text. We have therefore removed the attack success rate we added for text. We have also updated the text examples we initially gave. Furthermore, we have carried out a new manual evaluation on samples that are adversarial both for our method and Zhao et al. (2018b).

Reviewer 2: Thanks for engaging with us. We have fixed the typos and put the algorithm in one page. We have also changed the terminology ``adversarial success rate'' to ``attack success rate''. We also understand your point of view. In light of the changes we brought to the manuscript, we hope you will favourably rate our paper.

Reviewer 3: We wish you engaged with us like the other reviewers. We have updated the section ``Manifold Learning via SVGD'' a bit to highlight why a Bayesian framework was needed. In substance, we want to generate semantics preserving adversarial examples. In order to achieve this goal, we need to characterize as faithfully as possible the semantics of our inputs. That is, we need to learn the manifold that captures the semantics of the inputs. We use a VAE for manifold learning. Typically, VAEs learn an encoding function that maps the data manifold to an isotropic Gaussian, and a decoding function that transforms a latent code back to a sample in the input space. The reason for choosing the isotropic Gaussian is to render the ELBO tractable; hence easy to optimize. However, such a constraint imposed on the data manifold leads to learning poorly the semantics of the data, as reported by Jimenez Rezende \& Mohamed (2015). Given that we want to generate semantics preserving adversarial examples, it is important that we characterize the semantics of the inputs well and with minimal assumptions about the distribution of the data. In that regard, similar to Pu et al. (2017), we propose to use SVGD. With SVGD, we don't need to impose a parametric/functional form for the distribution we want to learn. As SVGD is an MCMC method, we would, however, need to capture the uncertainty that results from the approximations. Kim et al. (2018) have shown that SVGD can be cast as a Bayesian approach for parameter estimation and uncertainty quantification. Hence, the motivation for using a Bayesian framework.

We are *not* using another recognition network for manifold alignment. To clarify, we have the encoder $E'$ that perturbs the latent codes generated by the encoder $E$. Like $E$, $E'$ is parameterized by $M$ model instances. For large $M$, maintaining $M$ models is computationally intensive. Thus, we introduced the recognition network $f_\eta'$ to sample such models. The manifold alignment is a regularization technique that constrains $E'$ to follow a distribution similar to $E$.

With respect to the approximations, it is a wide practice to use empirical risk minimization to approximate distributions. While such an approximation can sometimes be problematic, we have shown that the distribution that $E'$ follows is similar to the distribution of $E$. We showed in the Appendix in Fig 5 that these distributions overlap quite well for most of the datasets we considered. Also, using toy data, we showed that the manifolds of the benign examples and the adversarial ones overlap quite well. Finally, the quality of the adversarial images is quite high compared to the baselines we considered.

For the reconstruction loss, in Equation 6, as a reminder, this loss is part-and-parcel of the training of VAEs. Although Song et al. (2018) and Zhao et al. (2018b) did not constrain the distance between adversarial examples and corresponding inputs, both use GANs which approximate the true data distribution. As a result, we would expect the adversarial examples they generate to be as realistically looking as the clean inputs. However, this is not the case. Also, note that Song et al. (2018) use AC-GAN with gradient penalty (an $L_2$ norm) to constrain their discriminator to lie within 1-Lipschitz functions. Finally, standard VAEs (with an $L_2$ reconstruction loss) are well known to generate blurry images of poor quality. We don't therefore think there is any competitive advantage the reconstruction loss brings to us that the gradient penalty doesn't confer to them.

Regarding the use of $L_\infty$ as reconstruction loss, now we attack the certified defences using this loss. We have run new experiments using this loss, and we have updated the paper.

---

> ### Comment · AnonReviewer1 · 2019-11-14
> **Updated review**
>
> I'm not sure reviewers and authors get notified if I just edit my original review text.  I'm posting a new comment here to note that I have updated my review after having seen the response and revision.  My concerns about the NLP attacks being wrong have been mostly addressed, and I raised my score to a 6 (noting that I am not evaluating the theoretical merits of the method presented).

---

> > ### Author Response · Authors · 2019-11-15
> > **comment**
> >
> > Thanks again for the insightful comment. We have added some examples that our model wrongly classified as adversarial in Table 9. We have more examples of real adversarial examples and real misses we would be happy to share.

---

### Author Response · Authors · 2019-11-15
**Updated paper with examples wrongly classified by the model as adversarial**

We thank Reviewer 1 once again for the insightful comment. Indeed, incorporating only adversarial examples sounded a lot like we have a 100% success rate but that's not the case. This was an oversight. Thus, we have added Table 9 in the appendix to show some examples that our approach considered as adversarial and that aren't after manual evaluation.

---

### Decision · Program_Chairs · 2019-12-19

**Decision:**

Reject

**Comment:**

This paper describes a method for generating adversarial examples from images and text such that they maintain the semantics of the input.

The reviewers saw a lot of value in this work, but also some flaws.  The review process seemed to help answer many questions, but a few remain: there are some questions about the strength of the empirical results on text after the author's updates. Wether the adversarial images stay on the manifold is questioned (are blurry or otherwise noisy images "on manifold"?).  One reviewer raises good questions about the soundness of the comparison to the Song paper.

I think this review process has been very productive, and I hope the authors will agree.  I hope this feedback helps them to improve their paper.